# Constitutive Damage Model of Foamed Lightweight Concrete Using Statistical Damage Theory

**DOI:** 10.3390/ma16175946

**Published:** 2023-08-30

**Authors:** Zhong Zhou, Yidi Zheng, Guiqiu Xie, Fan Li, Zigang Ji, Chenjie Gong

**Affiliations:** 1School of Civil Engineering, Central South University, Changsha 410075, China; 2Southwest Company of China Communications Construction, Chengdu 610041, China

**Keywords:** foamed lightweight concrete, statistical damage theory, constitutive model, pore damage

## Abstract

Foamed lightweight concrete has been applied in different fields of civil engineering because of its superior properties, but the related research considering internal pore damage is limited. Based on statistical damage theory and considering the uneven distribution of fracture damage and strength between the pores of light concrete, a damage constitutive model of foamed lightweight concrete was established based on the Weibull function. The parameters of the damage model were determined through a triaxial compression test, and the rationality was verified by combining the existing test data. Comparative tests show that the theoretical calculation results of the proposed statistical damage model of foamed light soil are consistent with the general trend of the experimental results, reflecting the value of the peak stress and strain and describing the overall development law of the stress and strain. The best fit was obtained when the confining pressure was 0.3 MPa and the density was 700 kg·m^−3^. The suggested damage constitutive method is highly applicable, which is of great significance to the microscopic mechanical analysis of foamed light concrete and the structural design in civil engineering.

## 1. Introduction

Foamed lightweight concrete is an excellent cementitious material with the advantages of low bulk capacity, high strength, low thermal conductivity, and sound insulation and has been widely used in construction and transportation [1,2,3]. The special pore structure of the foamed lightweight concrete makes its mechanical behavior different from that of ordinary concrete, so it is of great significance to study its damage characteristics for structural safety [4,5,6].

The microstructure of concrete affects the stress distribution. Cracks will be caused between the foam voids under the action of load, which will affect the structural strength [7,8]. To intensively elucidate this effect, many scholars have carried out relevant tests on the constitutive model of foam lightweight concrete. Su et al. [9] developed a theoretical model with 10 different damage factors based on a large number of compression tests to predict the nonlinear deformation of foam lightweight concrete under large loads and, on this basis, proposed an elliptical phenomenological yield criterion. Liu et al. [10] improved the Zhu–Wang–Tang model based on the impact properties of concrete, making it applicable to the stress–strain curve of foamed lightweight concrete at various loading rates. Guo et al. [11] studied the uniaxial compression parameters of foamed lightweight concrete with respect to temperature and strain rate and constructed a corresponding nonlinear mechanical calculation model using Lemaitre’s principle. Luo et al. [12] conducted compression experiments on porous concrete, on the basis of which they developed a static theory to study the constitutive relationship of foam concrete. Zhou et al. [13] constructed a predictive model for the concrete elastic modulus based on the spherical pore assumption and Walsh’s theoretical formulation and validated it through compression tests. However, relevant studies considering the pore damage inside foam concrete are limited.

Many new methods are applied to the study of concrete microstructure with the development of technology. Batool and Bindiganavile [14] combined CT scanning technology to study the effects of thermal conductivity, density, and volcanic ash admixture on the porosity of foamed concrete. CT scanning can accurately obtain the microstructure of cement, but it relies on high-performance equipment. In addition, deep learning is also predicted for the strength of foamed concrete. Nguyen et al. [15] predicted the compressive strength of foamed lightweight soil based on deep learning. This deep neural network improves the prediction accuracy through multiple hidden layers. However, this model is too dependent on data, and the training results are difficult to apply to other situations. Falliano et al. [16] obtained the cracking path of foamed concrete by field emission scanning electron microscopy, and calculated the fracture energy by the displacement of the crack opening. Yang et al. [17] added multi-scale fibers to foamed lightweight concrete, and the prepared material showed excellent performance in uniaxial compression tests.

The statistical damage theory of rock and conventional concrete provides a useful reference for the analysis of lightweight concrete. Su [18] constructed an elastoplastic damage constitutive model for foamed lightweight concrete on the basis of the principles of irreversible thermodynamics, continuum damage mechanics, and plasticity mechanics, and this model considers the effects of plastic deformation and damage evolution. Dara et al. [19] investigated internal cracks in the material caused by energy absorption, which are similar to pore damage inside foamed lightweight concrete. In addition, the Weibull probability distribution function provides a useful reference for damage research of foamed light soil. Kavussi et al. [20] applied the Weibull function to the damage study of materials and determined the possibility of the cracking failure of recycled asphalt pavement. Adamu et al. [21] studied the impact resistance of roller-compacted concrete and found that the crack distribution of concrete materials follows a two-parameter Weibull distribution function. However, the concrete they mainly studied did not involve foam additives, and insufficient consideration was given to internal pore changes. These studies illustrate the feasibility of using statistical damage theory and the Weibull function to study the stress–strain relationship in lightweight concrete, but relatively few research results are available on the compression damage theory of foam lightweight concrete. In addition, traditional methods are usually based on empirical formulations and simplifying assumptions, which may lead to discrepancies between design results and actual behavior. In contrast, the proposed model takes into account the complexity of internal pore damage and strength distribution, which can more accurately capture the damage behavior of the material and, thus, improve the performance of the structure. Therefore, the fracture damage variables of lightweight concrete should be defined, and a damage constitutive model for lightweight concrete considering fracture damage should be proposed.

In this study, the fracture damage and strength distribution between the pores of foam lightweight concrete are considered, and a statistical damage constitutive model of foamed lightweight concrete is established on the basis of statistical damage theory to intensively study the fracture damage evolution process of foam lightweight concrete.

## 2. Application of Foamed Lightweight Concrete

Foamed lightweight concrete is a lightweight cement-based material formed by preparing an aqueous solution of foaming agent into foam; mixing it with water, cement, additives, and other materials at a specific ratio; and hardening it. The preparation is shown in Figure 1. The transformation of the foam into pores after curing the cement effectively reduces the self-weight of the foam lightweight concrete, endowing the material with various physical and mechanical characteristics, such as light weight, adjustable density and strength, high mobility, incompressibility, and high strength.

Foam lightweight concrete can be applied to many building structures and foundation projects in practical engineering by mixing on-site, spraying, or making prefabricated blocks. As shown in Figure 2, foam lightweight concrete is used as the roadbed in the Guangzhou-Lianzhou expressway. Foam lightweight concrete can replace conventional concrete in filling foundations, effectively reducing loads, filling settlement areas and potholes, and improving the stability of the soil layer. In underground projects, structural safety is often limited by defects [22], but foam lightweight concrete offers a new solution, in addition to timely detection and repair. Foam lightweight concrete can be used as the filling material for underground tunnels and pipeline protection and to reduce the load and heat conduction of underground structures and improve the safety and reliability of the projects.

## 3. Damage Constitutive Model

### 3.1. Statistical Damage Variable

Many initial pores exist inside foam lightweight concrete, and the nature of its damage process is the damage evolution process of the compression, closure, and connectivity of internal bubble pores. Therefore, the fine structure of foam lightweight concrete can be generalized into three parts: the lightweight concrete matrix, the pores, and the fracture channels existing between pores, as shown in Figure 3. When the bubble pores inside foam lightweight concrete are intact and independent, the individual pores have a certain strength. After compression deformation occurs, fracture channels are created between pores. Small intact pores gradually develop into large pores, which leads to the loss of deformation properties within the lightweight concrete. Defining this fracture loss is the key factor.

The notations involved in this paper are summarized in Table 1. 

When lightweight concrete is compressed, numerous inter-pore fracture channels are generated internally and develop gradually with the increasing strain, leading to a rapid reduction in the elastic modulus of lightweight concrete. Therefore, the fracture damage variable of foam lightweight concrete can be defined as shown in Equation (1):(1)$$D_{c}=1-\frac{E_{c}}{E_{0}},$$

### 3.2. Weibull Distribution Function

The bubble pores inside the foam lightweight concrete are irregular and randomly distributed, while the lightweight concrete element contains many microscopic cracks and fissures, with varied strength values. The Weibull-based probability distribution function is easy to integrate, and its mean and range of values are greater than 0, which satisfies the features of foam lightweight concrete damage [23]. Therefore, the micro-element strength of the foam lightweight concrete is assumed to obey the Weibull function, as shown in Equation (2):(2)$$P(f)=\left\{ \begin{array}{cc} \frac{m}{K}{\left(\frac{f}{K}\right)}^{m-1}\exp\left[-{\left(\frac{f}{K}\right)}^{m}\right], & f\geq 0\\ 0, & f<0 \end{array}\right.\;\;\;.$$

The Weibull parameter of the foam lightweight concrete in the fracture damage state is expressed as Equation (3):(3)$$\begin{array}{l} m(C)=m_{0}(1-D_{c})\\ K(C)=K_{0}(1-D_{c}) \end{array}\},$$
where *m*_0_ and *K*_0_ are the Weibull parameters of the foam lightweight concrete in the lossless pore state, and *m*(*C*) and *K*(*C*) are the Weibull parameters of the foam lightweight concrete in the fracture damage state.

### 3.3. Derivation of Damage Constitutive Model

In this study, the total damage of the foam lightweight concrete is determined as the ratio of micro-elements to the total damage items at a certain stress state, as in Equation (4):(4)$$D=\frac{N_{f}}{N},$$
where *N_f_* and *N* are the number of micro-elements destroyed and the total number, respectively.

According to statistical damage theory [18], the micro-element strength of the foam lightweight concrete obeys a statistical distribution. Under the continuous loading of axial stress, the effective micro-elements inside the lightweight concrete are continuously destroyed and transformed into failed micro-elements. The number of failed micro-elements is calculated as shown in Equation (5). The damage ratio *D* can be obtained as in Equation (6).
(5)$$N_{f}=\int_{0}^{f}NP(f)d\sigma .$$
(6)$$D=1-\exp\left[-{\left(\frac{f}{K_{0}(1-D_{c})}\right)}^{m_{0}(1-D_{c})}\right].$$

In the case of axial compression, the relationship between strain and stress in the undamaged part of the lightweight concrete can be obtained according to the generalized Hooke’s law. The stress equation considering damage is organized to obtain Equation (7):(7)$$\sigma_{i}=E\varepsilon_{i}\times \exp\left[-{\left(\frac{f}{K_{0}(1-D_{c})}\right)}^{m_{0}(1-D_{c})}\right]+v\left(\sigma_{j}+\sigma_{k}\right),\;(i,j,k=1,2,3).$$
where *σ* = (1 – *D*)*σ*′.

The damage evolution equation for the foam lightweight concrete considering fracture damage is shown in Equation (8):(8)$$\sigma_{i}=E_{0}(1-D_{c})\varepsilon_{i}\times \exp\left[-{\left(\frac{f}{K_{0}(1-D_{c})}\right)}^{m_{0}(1-D_{c})}\right]+v\left(\sigma_{j}+\sigma_{k}\right),\;(i,j,k=1,2,3).$$

The Mohr–Coulomb criterion (hereafter referred to as the M-C criterion) [24] is introduced for lightweight concrete with the strength theoretical function. The expression for the strength of the foam lightweight concrete micro-element is shown in Equation (9):(9)$$f=\frac{E\varepsilon_{1}}{\sigma_{1}-v(\sigma_{2}+\sigma_{3})}\left[(\sigma_{1}-\sigma_{3})-(\sigma_{1}+\sigma_{3})\sin\varphi \right]-2c\cos\varphi .$$
where *c* is the cohesive force of the foam lightweight concrete, and *φ* is the internal friction angle of the foam lightweight concrete.

When the stress level is lower than the critical cracking stress, the fissures will not expand significantly and can be regarded as a non-damaged state. Thus, this critical stress can be described by threshold stress *σ_c_*, and the corresponding strain is threshold strain *ε_c_*. The damage constitutive model of the foam lightweight concrete in the damage state (after the damage threshold) is shown in Equation (10):(10)$$\sigma_{i}=\bar{A}\varepsilon_{i}{}^{2}+\bar{B}\varepsilon_{i}.$$

Combining Equations (8) and (10), the statistical damage constitutive model of foam lightweight concrete with fracture damage is proposed as Equation (11):(11)$$\sigma_{i}=\left\{ \begin{array}{l} \;\;\bar{A}\varepsilon_{i}{}^{2}+\bar{B}\varepsilon_{i},\;\;\;0\leq \varepsilon \leq \varepsilon_{c}\\ E_{0}(1-D_{c})\varepsilon_{i}\times \exp\left[-{\left(\frac{f}{K_{0}(1-D_{c})}\right)}^{m_{0}(1-D_{c})}\right]+v\left(\sigma_{j}+\sigma_{k}\right),\;\;\;\varepsilon >\varepsilon_{c} \end{array}\right..$$

### 3.4. Determination of Damage Model Parameters

Parameters *E*_0_, *v*, *c*, and *φ* in the proposed statistical damage constitutive model for foam lightweight concrete are relatively common and can be obtained based on conventional uniaxial or triaxial compression tests. The internal nominal stresses (*σ*_1_, *σ*_2_, *σ*_3_) and nominal strain (*ε*_1_) in lightweight concrete can also be obtained through compression tests. Therefore, the parameters $\bar{A}$, $\bar{B}$, *m*, and *K* in the damage model need to be determined.

#### 3.4.1. Weibull Parameters m and K

Equation (12) can be obtained from Equation (11):(12)$$\sigma_{1}=E\varepsilon_{1}\cdot \exp\left[-{\left(\frac{f}{K}\right)}^{m}\right]+2v\sigma_{3},$$
where *f* is calculated by Equation (13):(13)$$f=\frac{E\varepsilon_{1}}{\sigma_{1}-2v\sigma_{3}}\left[(\sigma_{1}-\sigma_{3})-(\sigma_{1}+\sigma_{3})\sin\varphi \right]-2c\cos\varphi .$$

During triaxial compression, peak stress *σ_p_* and peak strain *ε_p_* exist in the stress–strain relationship of the foam lightweight concrete. The first derivative of stress with respect to strain at this position is zero, as shown in Equation (14):(14)$${\frac{\partial \sigma }{\partial \varepsilon }|}_{\begin{array}{l} \sigma =\sigma_{p}\\ \varepsilon =\varepsilon_{p} \end{array}}=E\times \exp\left[-{\left(\frac{f(\sigma_{p})}{K}\right)}^{m}\right]\left\{1-\varepsilon_{p}{\left(\frac{f(\sigma_{p})}{K}\right)}^{m-1}\frac{mE\left[(\sigma_{p}-\sigma_{3})-(\sigma_{p}+\sigma_{3})\sin\varphi \right]}{K(\sigma_{p}-2v\sigma_{3})}\right\}=0.$$

The computational equations for *m* and *K* can be obtained by combining Equation (14), as shown in Equation (15):(15)$$\left\{ \begin{array}{l} m=\frac{f(\sigma_{p})(\sigma_{p}-2v\sigma_{3})}{E\varepsilon_{p}\left[(\sigma_{p}-\sigma_{3})-(\sigma_{p}+\sigma_{3})\sin\varphi \right]\ln\left[E\varepsilon_{p}/(\sigma_{p}-2v\sigma_{3})\right]}\\ K=\frac{f(\sigma_{p})}{\sqrt[{m}]{\ln\left[E\varepsilon_{p}/(\sigma_{p}-2v\sigma_{3})\right]}} \end{array}\right.,$$
where
(16)$$f(\sigma_{p})=\frac{E\varepsilon_{p}}{\sigma_{p}-2v\sigma_{3}}\left[(\sigma_{p}-\sigma_{3})-(\sigma_{p}+\sigma_{3})\sin\varphi \right]-2c\cos\varphi .$$

#### 3.4.2. Parameters $\bar{A}$ and $\bar{B}$

The constitutive curves for the compression deformation of foam lightweight concrete have continuous stress values at the threshold stress, and the first derivatives of stress with respect to strain remain consistent, as shown in Equations (17) and (18):(17)$$\bar{A}\varepsilon_{c}{}^{2}+\bar{B}\varepsilon_{c}=E\varepsilon_{c}\times \exp\left[-{\left(\frac{f(\sigma_{c})}{K}\right)}^{m}\right]+2v\sigma_{3},$$
(18)$$2\bar{A}\varepsilon_{c}+\bar{B}=E\times \exp\left[-{\left(\frac{f(\sigma_{c})}{K}\right)}^{m}\right]\left\{1-\varepsilon_{c}{\left(\frac{f(\sigma_{c})}{K}\right)}^{m-1}\frac{mE\left[(\sigma_{c}-\sigma_{3})-(\sigma_{c}+\sigma_{3})\sin\varphi \right]}{K(\sigma_{c}-2v\sigma_{3})}\right\}.$$

Equation (19) for $\bar{A}$ and $\bar{B}$ can be obtained as:(19)$$\left\{ \begin{array}{l} \bar{A}=-E\times \exp\left[-{\left(\frac{f(\sigma_{c})}{K}\right)}^{m}\right]{\left(\frac{f(\sigma_{c})}{K}\right)}^{m-1}\frac{mE\left[(\sigma_{c}-\sigma_{3})-(\sigma_{c}+\sigma_{3})\sin\varphi \right]}{K(\sigma_{c}-2v\sigma_{3})}-\frac{2v\sigma_{3}}{\varepsilon_{c}^{2}}\\ \bar{B}=E\times \exp\left[-{\left(\frac{f(\sigma_{c})}{K}\right)}^{m}\right]\left\{1+\varepsilon_{c}{\left(\frac{f(\sigma_{c})}{K}\right)}^{m-1}\frac{mE\left[(\sigma_{c}-\sigma_{3})-(\sigma_{c}+\sigma_{3})\sin\varphi \right]}{K(\sigma_{c}-2v\sigma_{3})}\right\}+\frac{4v\sigma_{3}}{\varepsilon_{c}} \end{array}\right.,$$
where
(20)$$f(\sigma_{c})=\frac{E\varepsilon_{c}}{\sigma_{c}-2v\sigma_{3}}\left[(\sigma_{c}-\sigma_{3})-(\sigma_{c}+\sigma_{3})\sin\varphi \right]-2c\cos\varphi .$$

The related parameters have been determined in the constitutive model of statistical damage theory for foam lightweight concrete. The model can be further validated and analyzed by combining specific triaxial compression test data of lightweight concrete.

## 4. Model Validation

Given the difficulty of detecting the internal fracture damage of foam lightweight concrete by conventional methods, the degree of damage is difficult to quantitatively describe. Therefore, the reasonableness of the proposed statistical damage model for lightweight concrete can be verified by example analysis for foam lightweight concrete under fissure-free damage conditions. 

### 4.1. Comparison of Different Densities

To verify the proposed statistical damage theory, the existing data under different density conditions [24] were selected. The relevant parameters of this triaxial test are shown in Table 2 and Table 3.

The data are substituted into the theoretical model to obtain the computational parameters of the foam lightweight concrete constitutive model under different densities. Then, the stress–strain theoretical curves are plotted and compared with the experimental results, as in Figure 4. 

The results show that the theoretical results of the damage constitutive model show an overall consistent trend with the experimental data of Yuan et al. [24], reflecting the numerical magnitude of the peak stresses and strains and describing the overall development pattern of stresses and strains in lightweight concrete. The test curve shows a rapid decrease in stress after reaching the peak stress, which is due to the macroscopic cracks within the lightweight concrete and the stress drop from the brittle damage under pressure. The brittle damage of foam lightweight concrete is random and sudden, and accurate prediction through theoretical constitutive models is difficult. Overall, the derived theory for lightweight concrete can be considered reasonable through the comparison of experimental data and can provide useful references for its physical and mechanical analysis.

With the decrease of material density, the model fits better. This phenomenon may be due to the fact that in the case of smaller density, the pore structure inside the foamed lightweight soil is more significant, which leads to the material being more prone to damage. The proposed damage model takes into account the damage of the pores and the heterogeneity of the strength distribution, which can more accurately simulate the response of the material in the case of small density.

### 4.2. Comparison of Different Confining Pressures

Triaxial compression test data [25] for powdered foam concrete specimens of 700 kg·m^−3^ density at 0.1, 0.2, 0.3, and 0.4 MPa peritectic pressure were selected for analysis. The test data are shown in Table 4, and the calculated results are presented in Table 5. The stress–strain relationships are plotted according to the constitutive model, as shown in Figure 5.

The test data show (Figure 5) that the theoretical curve in this study matches the test curve best when the surrounding pressure is 0.3 MPa. The peak stress and corresponding strain values are relatively close, and the overall change trend is near the actual change. It is possible that the pressure affected the internal pore structure of the foam concrete. The microstructure of concrete is relatively stable at 0.3 MPa, and its damage state is most consistent with the Weibull distribution.

The fitting effect is poor when the confining pressure is 0.1, 0.2, and 0.4 MPa. This may be because when the confining pressure is too small, a portion of the pores are closed, and the concrete has uneven deformation. When the confining pressure is too large, more severe deformation and fracture will occur inside the material, resulting in a poor fitting effect. But, the overall pattern of change is consistent. In addition, increasing the envelope pressure within a certain range can increase the maximum stress and critical strain of lightweight concrete by analyzing the effect of various confining pressures on the relationship curves of stress and strain. This phenomenon shows that a reasonable increase in the surrounding pressure can effectively limit the development of internal fractures and, thus, improve the compressive deformation performance.

## 5. Parameter Sensitivity Analysis

We performed a single-factor analysis of the constructed statistical damage model for foam lightweight concrete to evaluate the influence of each factor on the predictive effect of stress–strain relationships. The main damage parameters involved include elastic modulus of the material (*E*), Poisson’s ratio (*ν*), and internal friction force and angle (*c*, *φ*). We changed the values of the studied parameters in turn and fixed all other parameters. The sensitivity of the corresponding parameters is studied by observing the curve changes at different parameter values. The benchmark values of the parameters are as follows: elastic modulus *E*_0_ = 1011 MPa, Poisson’s ratio *ν*_0_ = 0.21, internal friction force *c*_0_ = 0.7 MPa, and internal friction angle *φ*_0_ = 18°.These values correspond to the compression test with *ρ =* 800 kg·m^−3^ in Figure 4b [24]. The coefficients (*i*) were set to 0.8, 0.9, 1.0, 1.1, and 1.2, and the test values (*E*_i_, *v*_i_, *c*_i_, and *φ*_i_) of the parameters were obtained by multiplying the reference values by the coefficients, as indicated in Table 6. The stress–strain relationships are shown in Figure 6.

The comparison results show that the change in a single parameter has a minimal effect on damage factors $\bar{A}$ and $\bar{B}$. The initial part in the stress–strain curve did not change significantly, nor did the maximum stress and the relative critical strain. The second half of the curve is more variable than the first part, suggesting that the model is sensitive to elastic modulus *E*, internal friction *c*, and internal friction angle *φ*. Appropriately narrowing these parameters (with a weight of 0.9) can improve the fit of the simulated curve to the actual stress–strain curve.

## 6. Conclusions

We analyzed the internal pore damage mechanism of foam lightweight concrete and proposed a damage constitutive model considering microstructural fragmentation based on the statistical damage theory. The influence of each parameter on the predicted values was analyzed with relevant triaxial test data. The following detailed conclusions can be extracted from this study:

A damage constitutive model of foamed lightweight concrete on the basis of statistical damage theory is proposed, fully considering the strength heterogeneity caused by the micro-pore damage of foamed lightweight concrete under external load.The key parameters of the damage model were deduced by combining the Weibull function and the M-C criterion, and the rationality of the damage constitutive model was determined.The triaxial compression test data verify that the damage constitutive method can fit the stress–strain curve of the foamed lightweight concrete under external load well, and the best results are obtained when the confining pressure is 0.3 MPa and the density is 700 kg·m^−3^. The model is sensitive to the friction angle and cohesion, and the optimal correction coefficient is 1.1.

Despite the ideal results of the study, the following deficiencies remain:Although the selected density and confining pressure are widely used in different types of structures and engineering, there will be more structural and load conditions in actual engineering. Therefore, future research can verify the accuracy and applicability of the model through data from more projects.Due to the equipment limitation, theoretical analysis is mainly used. It is a significant expansion direction to consider the fracture channel in combination with CT scanning technology.

## Figures and Tables

**Figure 1 materials-16-05946-f001:**
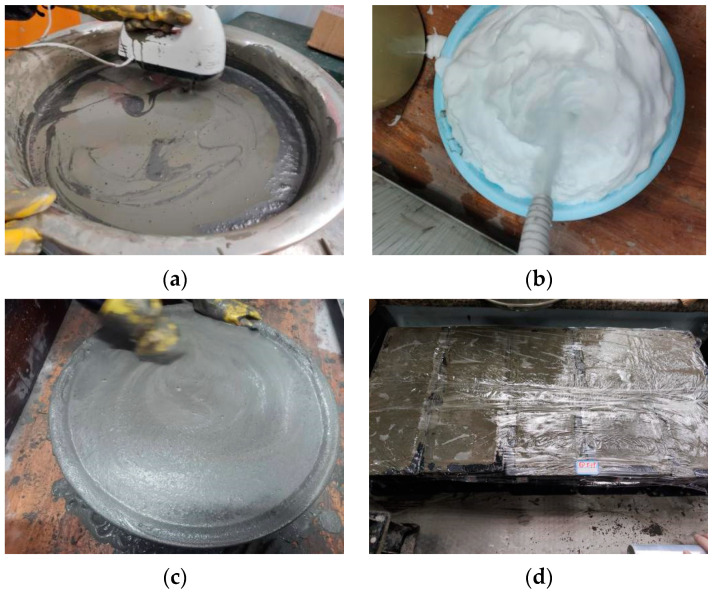
Production method of foamed lightweight concrete. (**a**) Cement slurry; (**b**) Foam; (**c**) Foam lightweight concrete (fluid state); (**d**) Foam lightweight concrete (test pieces).

**Figure 2 materials-16-05946-f002:**
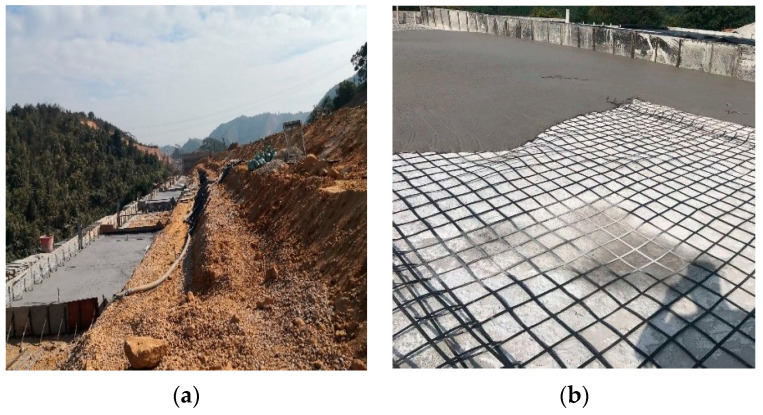
Application of foamed lightweight concrete in the Guangzhou-Lianzhou expressway. (**a**) Replacement of embankment; (**b**) Pouring of steel grating.

**Figure 3 materials-16-05946-f003:**
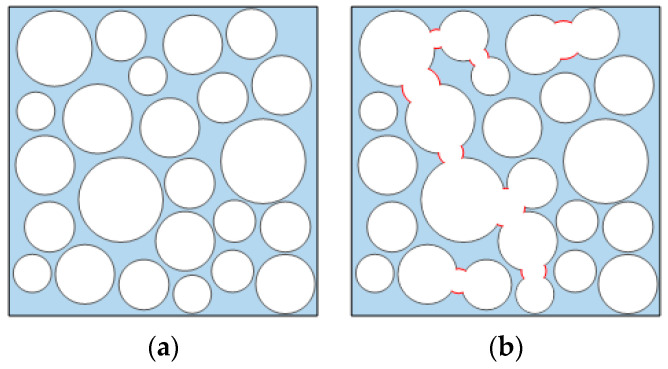
Schematic of foamed lightweight concrete mesoscopic model. (**a**) State of the initial pore space; (**b**) State of the fracture damage.

**Figure 4 materials-16-05946-f004:**
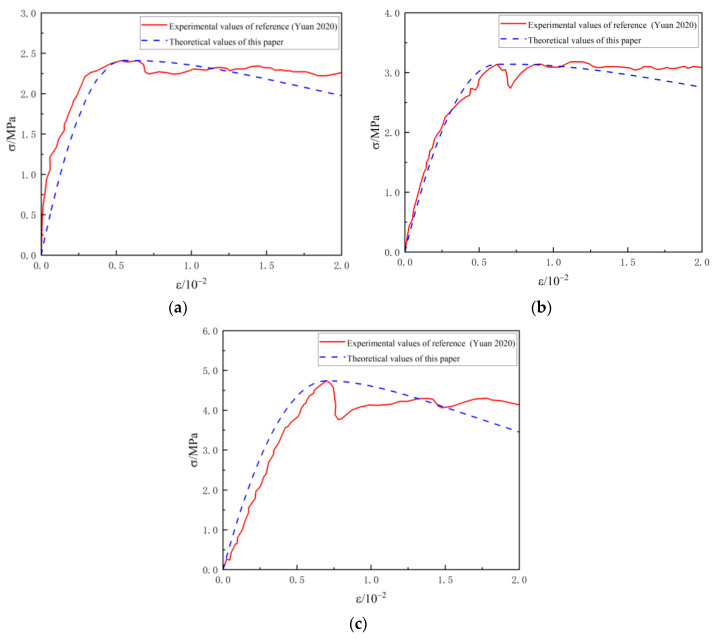
Comparison of relationship curves of stress and strain of lightweight concrete with different densities with reference [24]. (**a**) *ρ =* 700 kg·m^−3^; (**b**) *ρ =* 800 kg·m^−3^; (**c**) *ρ =* 900 kg·m^−3^.

**Figure 5 materials-16-05946-f005:**
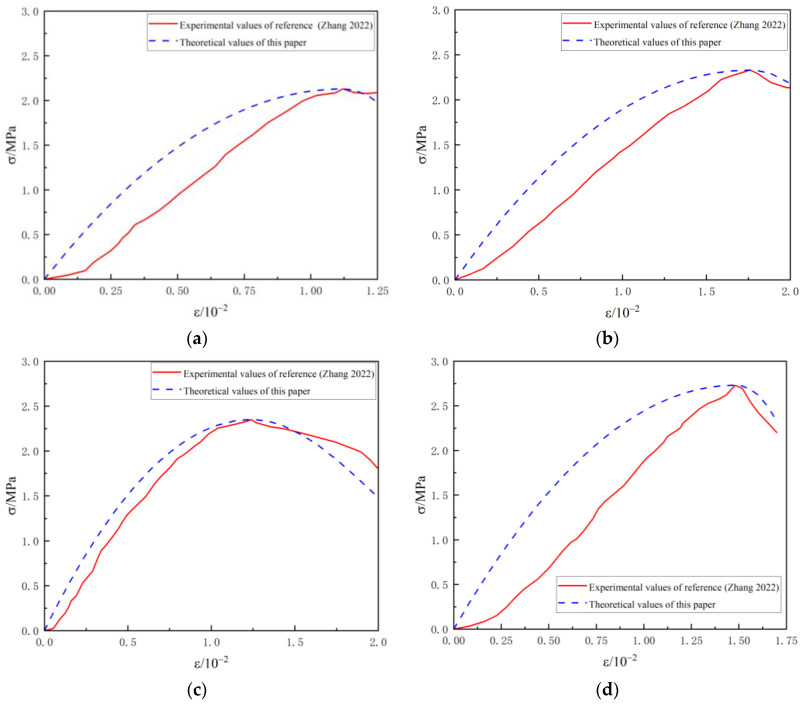
Comparison of relationship curves of stress and strain under various confining pressures with reference [25]. (**a**) *σ*_3_ = 0.1 MPa; (**b**) *σ*_3_ = 0.2 MPa; (**c**) *σ*_3_ = 0.3 MPa; (**d**) *σ*_3_ = 0.4 MPa.

**Figure 6 materials-16-05946-f006:**
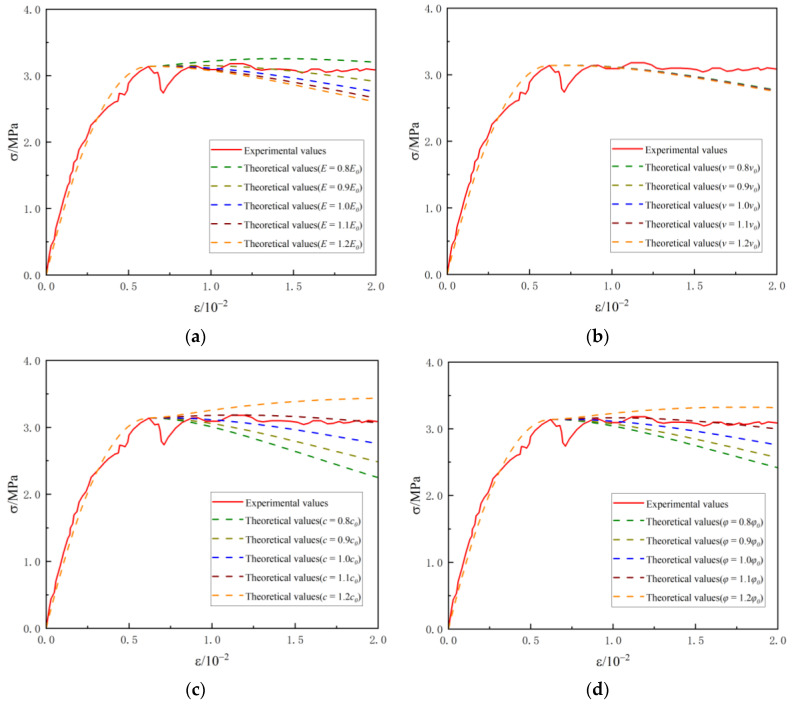
Prediction comparison of different parameters. (**a**) Elastic modulus *E*; (**b**) Poisson’s ratio *v*; (**c**) Internal friction force *c*; (**d**) Internal friction angle *φ*.

**Table 1 materials-16-05946-t001:** Basic notation summary.

Notation	Meaning
*D_c_*	Fracture damage variable
*E_c_*	Elasticity modulus after fracture damage
*E* _0_	Elasticity modulus before fracture is produced
*P*(*f*)	Probability density of Weibull distribution
*m*, *K*	Parameters of Weibull distribution
*σ*	Nominal stress
*σ_p_*	Peak stress
*ε*	Strain produced by the nominal stress acting on the damaged material
*ε*′	Strain produced by the effective stress acting on the undamaged material
*ε_p_*	Peak strain
$\bar{A}$, $\bar{B}$	Parameters of the damage model

**Table 2 materials-16-05946-t002:** Relevant parameters of foamed lightweight concrete triaxial test with various densities [24].

*ρ*/kg·m^−3^	*c*/Mpa	*φ/*(°)	*E*/Mpa	*σ_p_*/Mpa	*ε_p_*	*v*	*σ*_3_/Mpa
700	0.7	18	1011	2.41	0.0054	0.21	0.5
800	1.01	20	1217	3.14	0.0062	0.22	0.5
900	1.26	22	1460	4.74	0.0070	0.21	0.5

**Table 3 materials-16-05946-t003:** Related parameters of constitutive model with various densities.

*ρ*/kg·m^−3^	*m*	*K*	$\bar{A}$	$\bar{B}$
700	0.5164	1.4172	−82,647.4623	892.5926
800	0.4988	1.8947	−81,685.7440	1012.9032
900	0.6700	3.8099	−96,734.6939	1354.2857

**Table 4 materials-16-05946-t004:** Relevant parameters of foamed lightweight concrete triaxial test under different confining pressures [25].

*ρ*/kg·m^−3^	*c*/MPa	*φ/*(°)	*E*/MPa	*σ_p_*/MPa	*ε_p_*	*v*	*σ*_3_/MPa
700	0.65	23.9	208	2.13	0.0112	0.23	0.1
700	0.65	23.9	148	2.33	0.0176	0.23	0.2
700	0.65	23.9	255	2.35	0.0124	0.23	0.3
700	0.65	23.9	189	2.73	0.0148	0.23	0.4

**Table 5 materials-16-05946-t005:** Related calculation parameters of the proposed constitutive model with various confining pressures.

*σ*_3_/MPa	*m*	*K*	$\bar{A}$	$\bar{B}$
0.1	8.1386	5.4866	−16,980.2296	380.3571
0.2	6.0484	6.5911	−7521.9525	264.7727
0.3	2.6132	9.0041	−15,283.5588	379.0323
0.4	9.8381	6.8319	−12,463.4770	368.9189

**Table 6 materials-16-05946-t006:** Range of parameter values.

*i*	*E_i_*	*v_i_*	*c_i_*	*φ_i_*
0.8	*E*_1_ = 0.8 *E*_0_	*v*_1_ = 0.8 *v*_0_	*c*_1_ = 0.8 *c*_0_	*φ*_1_ = 0.8 *φ*_0_
0.9	*E*_2_ = 0.9 *E*_0_	*v*_2_ = 0.9 *v*_0_	*c*_2_ = 0.9 *c*_0_	*φ*_2_ = 0.9 *φ*_0_
1.0	*E*_3_ = 1.0 *E*_0_	*v*_3_ = 1.0 *v*_0_	*c*_3_ = 1.0 *c*_0_	*φ*_3_ = 1.0 *φ*_0_
1.1	*E*_4_ = 1.1 *E*_0_	*v*_4_ = 1.1 *v*_0_	*c*_4_ = 1.1 *c*_0_	*φ*_4_ = 1.1 *φ*_0_
1.2	*E*_5_ = 1.2 *E*_0_	*v*_5_ = 1.2 *v*_0_	*c*_5_ = 1.2 *c*_0_	*φ*_5_ = 1.2 *φ*_0_

## Data Availability

Not applicable.

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
