# Peer review of "Constitutive Damage Model of Foamed Lightweight Concrete Using Statistical Damage Theory"

_materials, 2023, doi:10.3390/ma16175946_

Round 1

Reviewer 1 Report (New Reviewer)

the paper can be accepted

Author Response

Please see our response in the attached word file.

Reviewer 2 Report (New Reviewer)

1. Please revise the abstract and the introduction section and provide quantitative information on the research's finding and those of the literature.

2. The result of the study are not compared to those of past research in this area. Thismethod of statistical analysis done has been around for decades at least. Please provide detailed comparisons for the findings.

3. The english language of the manuscript must be revised. It is not all academic writing.

4. Oddly, some 90% of the references are from a specific country. This is unethical and has to be changed. Please cite authors internationally and not your colleagues, etc.

5. Please increase the content of findings in the conclusion section and add quantitative data.

Must be revised.

Author Response

Please see our response in the attached word file.

Reviewer 3 Report (New Reviewer)

This research establishes a damage model for foamed lightweight concrete based on statistical theory, which is crucial for accurate structural design and analysis in civil engineering. Its application ensures safer lightweight construction, yet addressing real-world structural diversity and incorporating advanced imaging technologies for further accuracy remain challenges. I recommend minor revisions to address the enhancement of the work's quality.

1.      In what ways does the suggested damage constitutive method contribute to the microscopic mechanical analysis of foamed lightweight concrete and structural design in civil engineering?

2. Explain the rationale behind choosing the Weibull function as the basis for the damage model and its compatibility with the statistical damage theory?

3.      How were the triaxial compression test data combined with the existing test data to validate the rationality of the damage constitutive model?

4.      Discuss more the proposed damage constitutive model to address the uneven distribution of fracture damage and strength between the pores of light concrete? Kindly consider the following works:

a.       Numerical and experimental investigations of novel nature-inspired open lattice cellular structures for enhanced stiffness and specific energy absorption

b.      A novel nature-inspired 3D open lattice structure for specific energy absorption

c.       Characterization of penetrate and interpenetrate tessellated cellular lattice structures for energy absorption

5.      How does the proposed model address the dynamic nature of foamed lightweight concrete's behavior under different loads and conditions?

Author Response

Please see our response in the attached word file.

Round 2

Reviewer 2 Report (New Reviewer)

Manuscript can be accepted.

Manuscript can be accepted.

This manuscript is a resubmission of an earlier submission. The following is a list of the peer review reports and author responses from that submission.

Round 1

Reviewer 2 Report

The submitted Article with the Manuscript ID: "materials-2523560" and the title: "Constitutive damage model of foamed lightweight concrete using statistical damage theory" investigates the fracture damage and strength distribution between the pores of foam lightweight concrete. The study mainly focuses on lightweight foam concrete's fracture damage evolution process. The developed damage constitutive model of foam light concrete is based on the Weibull function; the parameters were determined through a triaxial compression test. Further, the model is verified experimentally using the data from existing triaxial compression tests from the literature. The work is comprehensive, and the paper is well-organized and well-written. However, some issues, questions, and clarifications should be amended, and the article needs improvement to reach the required scientific level. The following comments and suggestions are raised for the authors' reference:

1. Although some background information is provided in the introduction, the literature review seems shallow and relatively limited. Consequently, the following relevant topics are suggested to be considered since they might contribute to the promotion of the aims of this study:

(a) The critical factor that proves the novelty of the study are the chosen selected principles for contributing to the developed constitutive damage model. In accordance with the literature, the microstructure also affects the stress distribution in the matrix, and the apportionment of the voids caused by foam bubble pores is of great importance for understanding the crack path to the fracture surface. It is recommended to be commented on more extensively.

(b) The latest development of experimental and numerical investigation on the effect of voids caused by various origins like foam bubble pores, degradation because of fire, corrosion, etc., in reinforced concrete elements implementing a more complicated stress distribution, also focusing on damage investigation and monitoring with different Non-destructive evaluation techniques as verification of the numerical results.

(c) The utilization of non-destructive techniques that also consider the scattering of the fracture energy in the microstructure is of great importance for understanding the crack path during the distribution of damage in the fracture surface.

(d) How the randomly distributed voids caused by bubble pores inside the foam could affect the mechanical behavior accordingly. The recent developments of machine learning in fracture behavior and prediction of mechanical properties.

(e) Research significance and especially the subsequent impact of the presented study on the state of the practice needs to be adequately established. Could this method develop a reliability model able to evaluate the mechanical properties of foamed lightweight concrete elements more efficiently and less time-consuming than already established analytical methodologies? A systematic literature review based on the previous recommendations would help.

The following relative articles could be used as examples of relative articles that could support the issues above:

-"The reinforcement attributes of multi-scale hybrid fiber throughout the uniaxial compression of ultra-low-weight foamed cement-based composites," Construction and Building Materials, 2020.

-"Acoustic monitoring for the evaluation of concrete structures and materials," in Acoustic Emission and Related Non-Destructive Evaluation Techniques in the Fracture Mechanics of Concrete (Second Edition), 2021.

-"Smart cementitious sensors with nano-, micro-, and hybrid-modified reinforcement: mechanical and electrical properties," Sensors, 2023.

2. Some variables need to be well-presented in several equations. Thus, a notation list should be added.

3. The subsequent impact of the presented study on the state of the practice needs to be included. It should be mentioned how the proposed constitutive damage model could be helpful in structural design in the construction industry.

4. A reference is needed about the fracture channels that are created between the pores.

5. How many specimens were used for the experimental test data? Why have only three densities been tested? Why does the test data consist only of one project? What could be the accuracy of the model if different projects had enriched the experimental test data?

6. In Figure 4, an explanation is needed why more similarity can be observed between the theoretical and experimental curves for the smaller density of 700kg/m3.

7. Figure 5 should explain why the theoretical curve in this study matches the test curve best when the surrounding pressure is 0.3 MPa. The explanation about the envelope pressure should include, for example, why at 0,4 MPa, the worst fit match between theoretical and experimental curves appeared.

8. Citations of the photographs illustrated in Figure 2 should be added.

The English needs some polishing since the phraseology seems cumbersome in several places. The language and the overall style of the presentation needs revision. For example, “we can…”, “we have…”, “we selected…”, “we analyzed…”, etc. should be avoided.